# Mapping Hydrothermal Zoning Pattern of Porphyry Cu Deposit Using Absorption Feature Parameters Calculated from ASTER Data

**Mengjuan Wu** [1,2,3,4], **Kefa Zhou** [1,2,3,4], **Quan Wang** [5,6,*] **and Jinlin Wang** [1,2,3,4]

1   State Key Laboratory of Desert and Oasis Ecology, Xinjiang Institute of Ecology and Geography, Chinese Academy of Sciences, Urumqi 830011, China
2   Xinjiang Key Laboratory of Mineral Resources and Digital Geology, Chinese Academy of Sciences, Urumqi 830011, China
3   Xinjiang Research Centre for Mineral Resources, Chinese Academy of Sciences, Urumqi 830011, China
4   University of Chinese Academy of Sciences, Beijing 100049, China
5   Faculty of Agriculture, Shizuoka University, Shizuoka 422-8529, Japan
6   Research Institute of Green Science and Technology, Shizuoka University, Shizuoka 422-8529, Japan
*   Correspondence: wang.quan@shizuoka.ac.jp; Tel.: +81-54-2383683

**Abstract:** Identifying hydrothermal zoning pattern associated with porphyry copper deposit is important for indicating its economic potential. Traditional approaches like systematic sampling and conventional geological mapping are time-consuming and labor extensive, and with limitations for providing small scale information. Recent developments suggest that remote sensing is a powerful tool for mapping and interpreting the spatial pattern of porphyry Cu deposit. In this study, we integrated in situ spectral measurement taken at the Yudai copper deposit in the Kalatag district, northwestern China, information obtained by the Advanced Spaceborne Thermal Emission and Reflection Radiometer (ASTER), as well as the spectra of samples (hand-specimen) measured using an Analytical Spectral Device (ASD) FieldSpec4 high-resolution spectrometer in laboratory, to map the hydrothermal zoning pattern of the copper deposit. Results proved that the common statistical approaches, such as relative band depth and Principle Component Analysis (PCA), were unable to identify the pattern accurately. To address the difficulty, we introduced a curve-fitting technique for ASTER shortwave infrared data to simulate Al(OH)-bearing, Fe/Mg(OH)-bearing, and carbonate minerals absorption features, respectively. The results indicate that the absorption feature parameters can effectively locate the ore body inside the research region, suggesting the absorption feature parameters have great potentials to delineate hydrothermal zoning pattern of porphyry Cu deposit. We foresee the method being widely used in the future.

**Keywords:** porphyry Cu deposit; hydrothermal zoning pattern; absorption feature parameters; ASTER

## 1. Introduction

Porphyry Cu systems are defined as large volumes (10->100 km$^3$) of hydrothermally altered rock centered on porphyry Cu stocks, presently supplying nearly three quarters of the world's Cu, half the Mo, perhaps one fifth of the Au, most of the Re, and minor amounts of other metals (silver, palladium, tellurium, selenium, bismuth, zinc, and lead) [1]. Porphyry Cu deposits typically occur in association with hydrothermal alteration zoning patterns that, in general, comprise potassic, phyllic, and propylitic from the center to the outward [2], deploying recognizable spatial features. The mineralization is closely related to alteration, which is a sign of mineralization scale and enrichment degree of the ore body. Furthermore, it is well known [1,2] that the ore body is mainly distributed in quartz-phyllitization

or potassic alteration zone, the larger the size of the deposit, the stronger the alteration. Moreover, the better the zoning, the higher the mineralization enrichment [3]. Consequently, a clear understanding of the characteristics and spatial distribution pattern of hydrothermal alteration zoning pattern in porphyry Cu systems plays a critical role in locating a prospecting target.

The traditional method for identifying the alteration zone is based on the restoration of the original rock properties and is determined according to the type and intensity of the altered minerals and the contents of major elements (e.g., $SiO_2$, $K_2O$, $Na_2O$, $CaO$, $FeO$, $Fe_2O_3$, and so on), which is very time-consuming and labor extensive. The Yudai deposit, which is the research site of this study, detailed drill core logging, and petrographic study have been used to delineate the alteration/mineralization zones, and the whole-rock geochemical data for the Yudai quartz diorite porphyry have been studied to discuss petrogenesis [4]. Comparatively, a recently developed method of using statistics to process geochemical profile data provides relatively quickly and accurately delineate the alteration-mineralization range and zoning of porphyry body and surrounding rock for narrowing the prospecting target area [3]. However, such geologically based approaches, in general, require a large number of hand specimen and thin section identification, making the division of the alteration zone a more complicated task and hard to keep up with the needs of prospecting and evaluation.

On the other hand, the mineral assemblages in alteration zone deploy diagnostic spectral absorption features in the visible near-infrared and shortwave infrared wavelength regions, such as phyllosilicates, carbonates, sulfates, Mg, Fe-bearing, and OH-bearing minerals [5–9]. Accordingly, remote sensing data with sufficient spatial and spectral information are promising for identifying spatial distributions of mineral groups. Using multispectral data to extract alteration information in the exposed area of bedrock has already achieved good results in previous studies [10–13].

To date, most popularly applied optical satellite-borne data on identifying alteration zone are Landsat Thematic Mapper (TM), Enhanced Thematic Mapper Plus (ETM+), Advanced Land Imager (ALI), and Advanced Spaceborne Thermal Emission and Reflection Radiometer (ASTER) data, due preliminary to their relatively high spatial resolution and affordable cost. For example, Landsat Thematic Mapper/Enhanced Thematic Mapper+ (TM/ETM+) images have been used as a tool to identify hydroxyl-bearing minerals in the reconnaissance stages of porphyry copper/gold exploration [14,15]. While the Advanced Land Imager has six unique wavelength channels spanning the visible and near-infrared (400–1000 nm), which is especially useful for detecting iron minerals in porphyry copper deposits [16,17]. However, although the hydroxyl-bearing alteration minerals have been separated from unaltered surrounding rocks in TM/ETM+ image, the individual alteration minerals cannot be identified due to the limitation of spectral resolution in shortwave infrared (SWIR).

In comparison, ASTER data has a relatively narrow spectral resolution in the shortwave infrared region, which can facilitate highlighting the presence of spectral absorption characteristics of Al-O-H, Mg-O-H, Fe-O-H, Si-O-H, and $CO_3$ molecular bonds, and would make it superior to other multispectral data (including TM) in the absence of usable hyperspectral data. In detail, illite/muscovite, which is a representative alteration mineral in the phyllic zone, yields an intense Al-OH absorption feature centered at 2200 nm, coinciding with ASTER band 6 (2185–2225 nm). The mineral assemblages of the outer propylitic zone, including epidote, chlorite, and calcite, exhibit absorption features situated at 2350 nm, which coincide with ASTER band 8 (2295–2365 nm). As thus, ASTER data have been widely used for geological/structural mapping and ore minerals exploration, particularly for porphyry copper deposits [18–23].

Most popular methods of applying ASTER data on mapping the hydrothermal zoning pattern of porphyry Cu deposit include Principle Component Analysis (PCA), Band Ratios (BR), Spectral Angle Mapper (SAM), and Mixture-Tuned Matched-Filtering (MTMF) [20]. Most of these methods have been used for identifying individual alteration mineral, but seldom in determining the absolute contents (intensity) of individual minerals, a sign of mineralization enrichment, and ore body position. Furthermore, the mapped hydrothermal zoning patterns using those methods can be misleading when the representative alteration minerals are distributed in different alteration zones.

A previous study by Harald [24] had used the technique to map iron absorption feature parameters based on synthetic Sentinel-2 imagery and reported that the curve-fitting technique could approximate the absorption features of several minerals, such as beryl, bronzite, goethite, hematite, and jarosite. In this study, we followed the approach to use ASTER SWIR bands to reconstruct the absorption features at a hyperspectral resolution for mapping hydrothermal zoning pattern with feature parameters, including the wavelength position of maximum absorption, together with the feature depth, describing the absorption shapes of the 2200 nm and 2350 nm. Besides, we have also used the common statistical approaches, including Relative Absorption Band Depth (RBD) and Principal Component Analysis (PCA), to map the hydrothermal zoning pattern of porphyry Cu deposit, providing an intensive comparison.

## 2. Methodology

### 2.1. Research Site

We selected the newly discovered porphyry Cu deposit in Kalatage Copper Polymetallic Ore Cluster area in eastern Tianshan, Western China, as the research site (Figure 1). This site has distinctive characteristics of geology, mineralization, geochemistry, and wall rock alteration of typical porphyry Cu deposits [25], providing an ideal location for linking with remote sensing data.

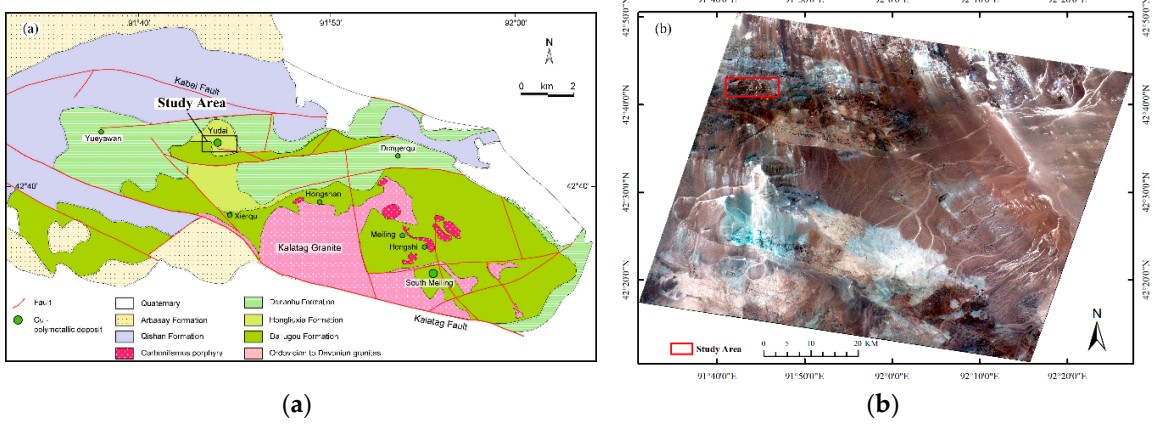

(**a**)                    (**b**)

**Figure 1.** (**a**) Regional geological map [26] of Kalatag district and (**b**) ASTER (Advanced Spaceborne Thermal Emission and Reflection Radiometer) image 3, 2, 1 in red, green, blue showing location of the Yudai porphyry copper deposit.

During the past ten years, several early Paleozoic Cu-polymetallic deposits have been discovered in the Kalatag district [27,28], including the South Meiling volcanogenic massive sulphide (VMS) deposit (comprised of the Honghai and Huangtupo ore blocks) [29,30] and Hongshi hydrothermal vein deposit [4], among which Yudai is the most recent [25]. Most deposits occur along with the belts, and the discovery of the Yudai has greatly expanded the exploration space in the eastern Tianshan, China.

The mineralization of the Yudai area is lenticular, vein-type, and disseminated in the porphyritic quartz diorite. The mineral association is represented by pyrite-chalcopyrite-magnetite (-chalcocite-molybdenite), and the gangue minerals include chlorite, quartz, and sericite with minor amounts of epidote and carbonate. The hydrothermal alteration is characterized by K-feldspar and biotite (potassic alteration), silica, chlorite, epidote, sericite, and carbonate [26]. An existing geological map in this region shows that the core of mineralized quartz diorite porphyry is surrounded by multiple alteration zones: the potassic + silication alteration zone, silication + sericitization alteration zone, and propylitization alteration zone. Although it is a typical porphyry type alteration zone, the propylitization zone appears in the silication + sericitization zone, causing difficulty in remote sensing information extraction. A range of alteration minerals, such as epidotization, chloritization,

peacock petrifaction, hematitization, and so on, are exposed on the surface. The major part of the study area has a well-exposed and sparse vegetated surface suitable for remote sensing investigations.

## 2.2. ASTER Data

ASTER has bands covering from visible near-infrared (520–860 nm), at a spatial resolution of 15 m, to shortwave infrared (1600–2430 nm), at a spatial resolution of 30 m, and to thermal region (8130–11650 nm), at a spatial resolution of 90 m [31]. The swath width of one ASTER scene is 60 km (each scene covers an area of $60 \times 60$ km$^2$) and is, hence, proper for regional mapping [32]. The ASTER data used in this study were level 1B acquired on 2 October 2001. The selected scene contains the research site with low cloud cover, which can provide spectral information about the lithology.

ASTER Level 1B data were converted to relative reflectance using the Flat Field method proposed by Roberts et al. [33] for reducing the atmospheric influences of the image and enhancing the spectral absorption characteristic of the surface material. Only shortwave infrared bands of ASTER data were used in this study. A subset scene of the ASTER used in this study is shown in Figure 1b, covering an area of 42.683–42.717 N, 91.683–91.783 E.

## 2.3. Informative ASTER Bands as Determined from Hyperspectral Information

Before image interpretation, we carried out extensive fieldworks in Yudai Ore for clarifying the occurrence and the spatial distribution of hydrothermal zoning pattern and the associated altered rocks types in this district as well. Samples (hand-specimen) representing the characteristic of mineral assemblages of different alteration zone and stratum were collected, as shown in Figure 2. Further, thin sections were prepared for microscopy to identify the alteration zone mineralogy of these collected samples.

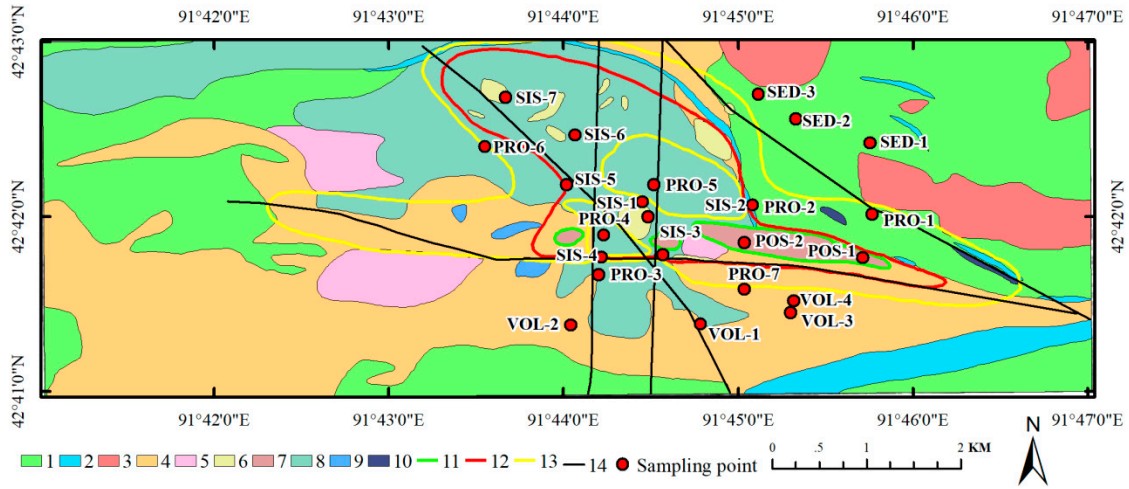

**Figure 2.** Geological map of the Yudai porphyry copper deposit (modified after [25]). 1-clastic sedimentary rock (D1d); 2-biogenic carbonates (D1d); 3-volcanic breccia; 4-dacitic volcanic and volcaniclastic rocks; 5-basalt; 6-pyrite felsite; 7-mineralized quartz diorite porphyry; 8-diorite porphyry; 9-gabbro intrusion; 10-siderite ore-bodies; 11-orebody, potassic + silication zone (POS); 12-silication + sericitization zone (SIS); 13-propylitization zone (PRO); 14-fault. Location of collected rock samples is denoted in the map with sample code.

The reflectance spectra of samples (hand-specimen) were measured using an Analytical Spectral Device (ASD) FieldSpec4 high-resolution spectrometer. ASD is designed to record the signal throughout the spectral region from 350 to 2500 nm with a sampling interval of 1.4–2 nm. In this study, spectral signatures were taken under a controlled dark environment in the laboratory. A tungsten filament halogen lamp with a wavelength range of 400 nm to 2500 nm was used as the artificial light

source for spectral data collection. The reflectance spectra were acquired nadir with the incidence angle setting to be 30°. All spectral measurements were made relative to a standard diffuse reflector. Five repeats were made for each time, and the standard calibration was carried out for every five samples. The data were internally interpolated to provide outputs at 1 nm intervals.

In this study, we only focused on the spectral ranges with known absorbance features for associated minerals, and corresponding ASTER SWIR bands were identified afterward (Table 1). Common absorption features were noted for all samples (hand-specimen) (Table 2). All spectra deployed very well defined 1900 nm hydration bands. Besides, samples showing absorptions around 2200 nm because of Al-OH vibrations suggested the existence of phyllosilicates minerals, such as kaolinite, illite, muscovite, etc. [18,34–36]. Meanwhile, the absorption features centered at around 2350 nm were indicative of carbonates/mafic minerals [18,34–36]. Most absorption features observed in hand-specimen from alteration zone and stratum did not suggest there was an apparent spatial zoning trend, for instance, the absorption feature around 2200 nm commonly appeared in silication alteration zone, propylitization alteration zone, and clastic sedimentary stratum. Similarly, the absorption feature around 2350 nm existed in all of the alteration zones. On the other hand, the positions of absorption features varied slightly in different hand-specimen, and different absorption band characteristics caused by the different alteration mineral contents contained in samples were noted as well (Table 2).

**Table 1.** Characteristic spectral features used for identification.

|  | **Characteristic Bands** |
|---|---|
| **Iron Oxides/Hydroxides** | |
| Jarosite | 2215 nm (doublet) |
| **Aluminosilicates** | |
| Illite/Kaolinite/Muscovite | 2200 nm |
| **Carbonates/Mafic Minerals** | |
| Calcite/Chlorite/Epidote | 2250 nm, 2350 nm |

**Table 2.** Characteristic absorption bands of laboratory reflectance spectra and their corresponding ASTER (Advanced Spaceborne Thermal Emission and Reflection Radiometer) bands, of the collected samples (hand-specimen) in the Yudai Cu deposit.

| Alteration Zone and Stratum | Sample No. | Rock Type | Absorption Band (nm) | Absorption Band Characteristics | Corresponding to ASTER Bands |
|---|---|---|---|---|---|
| Potassic + silication alteration zone | POS-1 | Dacite | 1918 | Very strong, narrow | |
| | | | 2211, 2350 | Moderate, narrow | Band 6, band 8 |
| | POS-2 | Potassium alteration dacite | 1933, 2350 | Strong, narrow | Band 8 |
| Silication + sericitization alteration zone | SIS-1 | Epidotization syenite | 1910, 2255 (doublet) | Moderate, narrow | Band 7 |
| | SIS-2 | Surface stalinization crust (brownish red) | 1937 | Strong, broad | |
| | | | 2210, 2435 | Weak, narrow | Band 6 |
| | SIS-3 | Limonitization powder | 1968 | Strong, broad | |
| | | | 2401 | Weak, narrow | Band 9 |
| | SIS-4 | Diabase | 1900, 2250, 2336 | Very weak, narrow | Band 7, band 8 |

**Table 2.** *Cont*.

| Alteration Zone and Stratum | Sample No. | Rock Type | Absorption Band (nm) | Absorption Band Characteristics | Corresponding to ASTER Bands |
|---|---|---|---|---|---|
| SIS-5 | | Surface stalinization crust (brownish red) | 1927 | Strong, broad | |
| | | | 2210 | Weak, narrow | Band 6 |
| | SIS-6 | Surface stalinization crust (brownish red) | 1927 | Moderate, broad | |
| | | | 2210 | Very weak, narrow | Band 6 |
| | SIS-7 | Surface stalinization crust (brownish red) | 1914, 2198 | Strong, narrow | Band 6 |
| Propylitization alteration zone | PRO-1 | Silicification rock | 1915 | Moderate, narrow | |
| | | | 2215 (doublet), 2385 | Weak, narrow | Band 6 |
| | PRO-2 | Covered gravel | Flat and straight, with no obvious absorption characteristics | | |
| | PRO-3 | Grey-green altered tuff | 1900 | Weak, narrow | |
| | | | 2255 (doublet) | Very strong, narrow | Band 7 |
| | PRO-4 | Grey-green altered almond-shaped andesite | 1950, 2261 (doublet) | Moderate, narrow | Band 7 |
| | PRO-5 | Yellow-green altered diorite porphyrite | 1947 | Moderate, narrow | |
| | | | 2258 (doublet) | Very strong, narrow | Band 7 |
| | PRO-6 | Grey-green tuff | 2183 (doublet) | Strong, narrow | Band 5 |
| | | | 1915 | Moderate, narrow | |
| | | | 2350 (doublet) | Weak, narrow | Band 8 |
| | PRO-7 | Light gray-brown syenite porphyry | Flat and straight, with no obvious absorption characteristics | | |
| Clastic sedimentary stratum (D1d) | SED-1 | Argillaceous rock | 1915 | Moderate, narrow | |
| | | | 2255 (doublet) | Moderate, narrow | Band 7 |
| | SED-2 | Diorite porphyrite | 1915 | Moderate, narrow | |
| | | | 2255 (doublet) | Moderate, narrow | Band 7 |
| | SED-3 | Dacite porphyry | 1911, 2215 | Strong, narrow | Band 6 |
| Dacitic volcanic and volcaniclastic stratum | VOL-1 | | | | |
| | VOL-2 | Flat and low reflectance (5%–25%), with no obvious absorption characteristics | | | |
| | VOL-3 | | | | |
| | VOL-4 | | | | |

*2.4. Absorption Feature Parameters for Mapping Hydrothermal Zoning Pattern*

Absorption feature parameters, such as the wavelength position, depth, and width, were applied to map the mineral spectral absorption, based on the assumption that the spectral data are continuous as by van der Meer [37]. As thus, we assumed that the ASTER SWIR data were continuous for mapping the mineral spectral absorption features. However, the ASTER data had wide gaps in SWIR (Table 3), requiring proper interpolation techniques to model the absorption feature shape from such spectrally

lower resolution bands. In this study, we fitted a quadratic polynomial function through the relative reflectance values of ASTER SWIR bands, which needs only 3 bands to determine the curve.

**Table 3.** Spectral position and bandwidth of ASTER (Advanced Spaceborne Thermal Emission and Reflection Radiometer) shortwave infrared (SWIR) bands.

| Band No. | Spectral Position ($\lambda$, nm) | Bandwidth ($\triangle \lambda$, nm) |
|:---:|:---:|:---:|
| 1 | 1656 | 10 |
| 2 | 2167 | 40 |
| 3 | 2209 | 40 |
| 4 | 2262 | 50 |
| 5 | 2336 | 70 |
| 6 | 2400 | 70 |

Specifically, the Al(OH)-bearing minerals deploy clear absorption features centered at 2200 nm [5], corresponding to ASTER band 6. On the other hand, the Fe/Mg(OH)-bearing and carbonates minerals exhibit absorption features situated at 2350 nm [4], coinciding with ASTER band 8. Accordingly, ASTER bands 5, 6, 7 and bands 7, 8, 9 were selected to chart the absorption feature parameters of Al(OH)-bearing, Fe/Mg(OH)-bearing, and carbonates minerals, respectively.

We applied a quadratic polynomial fitting method for wavelength positions $\lambda$ between 2145 nm and 2285 nm and 2235 nm and 2430 nm, respectively, at a 1 nm wavelength interval, to approximate the absorption feature at a hyperspectral resolution:

$$w_\lambda = m\lambda^2 + n\lambda + q, \tag{1}$$

where $w_\lambda$ is the interpolated reflectance value at position $\lambda$, $\lambda$ is the wavelength position in nm, and m, n, q are the coefficients of the quadratic polynomial function.

We examined the method on its feasibility using lab-taken spectra of hand-specimen, which had been resampled to ASTER bands using spectral response functions [38] of ASTER data. All samples collected from the alteration zone were used for examination.

The quadratic polynomial fitting method was then applied to image data. Pixels with negative quadratic polynomials or with positive quadratic polynomials with the point of maximum absorption outside the 2145–2285 nm in Al(OH)-bearing mineral or 2235–2430 nm wavelength range in Fe/Mg(OH)-bearing and carbonates minerals were excluded. For the remaining pixels passing through the screening, the wavelength of maximum absorption, as well as the depth of the feature, were determined. The interpolated wavelength positions of the minimum $w_{min}$ of the resulting quadratic polynomial function were calculated:

$$w_{min} = -n/2m, \tag{2}$$

where $w_{min}$ is the interpolated wavelength position with the minimum reflectance value.

On the other hand, the interpolated depth of that absorption feature $d_{min}$ was calculated as:

$$d_{min} = 1 - f(w_{min}). \tag{3}$$

*2.5. Statistical Approaches for Mapping the Hydrothermal Zoning Pattern*

2.5.1. Relative Absorption Band Depth (RBD)

Relative Absorption Band Depth (RBD) is a useful three-point ratio formulation for detecting diagnostic mineral absorption features [5]. For each absorption feature, the numerator is the sum of the bands representing the shoulders, and the denominator is the band located nearest the absorption feature minimum [5]. The relative absorption band depths of RBD 6 (band 6/[band 5 + band 7])

and RBD 8 (band 8/[band 7 + band 9]) were separately applied to extract Al(OH)-bearing minerals, Fe/Mg(OH)-bearing, and carbonates minerals. The results obtained with RBD 6 and RBD 8 did not display apparent zoning pattern, suggesting the RBD method is inadequate to distinguish alteration zone in this study area where one alteration zone appears in multiple absorption features (Figure 3).

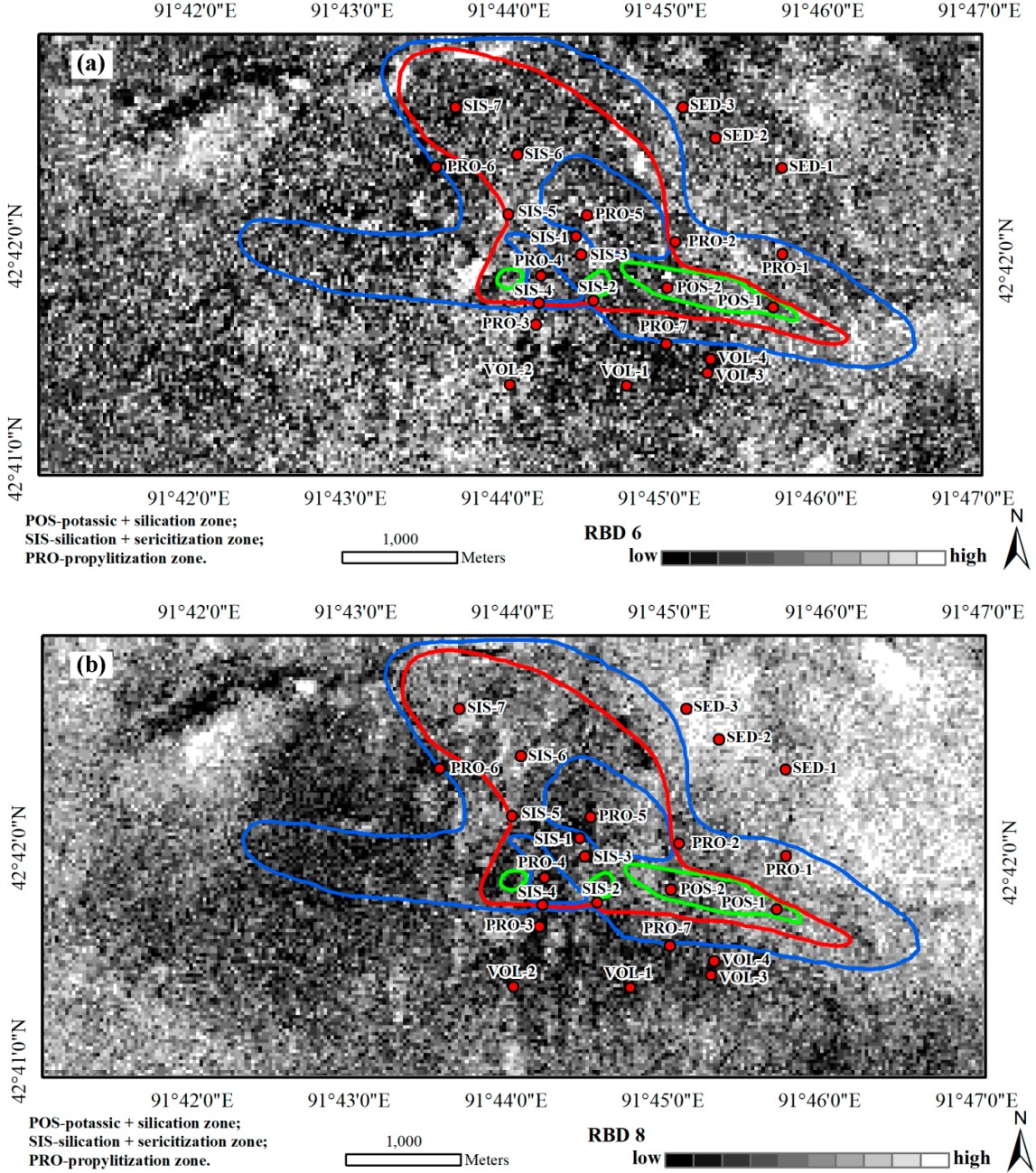

**Figure 3.** (**a**) RBD (Relative Absorption Band Depth) 6 [(band 5 + band 7)/band 6] and (**b**) RBD 8 [(band 7 + band 9)/band 8] overlaid with alteration zone and sampling point.

### 2.5.2. Principal Component Analysis (PCA)

PCA is a multivariate statistical method predicting which PC eigenvector matrix contained the target (mineral) information after applying PCA according to the magnitude and sign of the eigenvector loadings [6]. A subset of ASTER bands are the input data to reduce the dimension of data by avoiding the negative effect of unbeneficial bands to map the target material [39], while subsets were determined according to the position of characteristic spectral features of key alteration minerals. After applying

PCA, the eigenvector matrix used to calculate PCA for each subset was examined to identify which PC contained the target (mineral) information based on the criterion proposed by Loughlin [40].

We applied PCA to ASTER SWIR bands, and the outputs are presented in Table 4. According to Loughlin [40], a PC image with moderate-to-high eigenvector loadings for diagnostics reflective and absorption bands of mineral or mineral group with opposite signs enhances that mineral. According to spectral characteristics, ASTER band 4 covered the spectral region (~1600 nm) where all OH-bearing minerals had maximum reflectance values, while Al(OH)-bearing minerals, such as kaolinite, alunite, muscovite, and illite, showed major absorption features in ASTER bands 5, 6, 7 (2140–2280 nm). Furthermore, Fe/Mg(OH)-bearing minerals, such as chlorite, carbonates, such as calcite and dolomite, were well covered in the bands 8 and 9 (2290–2430 nm) of ASTER data. From the magnitude and the sign of the eigenvectors loadings, it is apparent that PC 3 and PC 4 contained the target information. PC 3 had high positive eigenvector loadings for the band 4 (0.382) and large negative loadings for the band 6 (−0.580). The PC 3 was supposed to discriminate the Al(OH)-rich rocks and hydrothermal alteration zones from their bright to dark gray signatures (Figure 4a). PC 4 had high positive eigenvector loadings for the band 4 (0.167) and large negative loadings for the band 8 (−0.396). It enabled the differentiation of the Fe/Mg(OH)-bearing, as well as carbonates minerals, by their bright image signature (Figure 4b).

**Table 4.** Eigenvector matrix of principal components (PC) analysis on ASTER (Advanced Spaceborne Thermal Emission and Reflection Radiometer) shortwave infrared (SWIR) bands.

|  | Band 4 | Band 5 | Band 6 | Band 7 | Band 8 | Band 9 |
|---|---|---|---|---|---|---|
| **PC1** | 0.457 | 0.388 | 0.420 | 0.411 | 0.436 | 0.326 |
| **PC2** | −0.782 | −0.108 | 0.006 | 0.197 | 0.437 | 0.384 |
| **PC3** | **0.382** | −0.521 | **−0.580** | 0.130 | 0.477 | 0.030 |
| **PC4** | **0.167** | −0.155 | −0.081 | −0.219 | **−0.396** | 0.859 |
| **PC5** | −0.081 | 0.425 | −0.575 | 0.605 | −0.338 | 0.037 |
| **PC6** | −0.003 | 0.601 | −0.387 | −0.602 | 0.348 | 0.080 |

### 2.6. Validation Data

In this study, the verification of hydrothermal zoning pattern interpreted from ASTER data was based on both field samples and the currently existing geological map. Field sampling was set to different strata and alteration zones (see locations in Figure 2). The sampling points in the alteration zone were well located in the area where the typical alteration minerals developed to ensure their representativeness. The number and rock type of each sample can be found in Table 2. Commonly, ASTER data is used for 1:50,000 scale geological mapping. As thus, the 1:50,000 scale geological mapping used in this study should have met with the requirements on validating the reliability of interpretation accuracy [41].

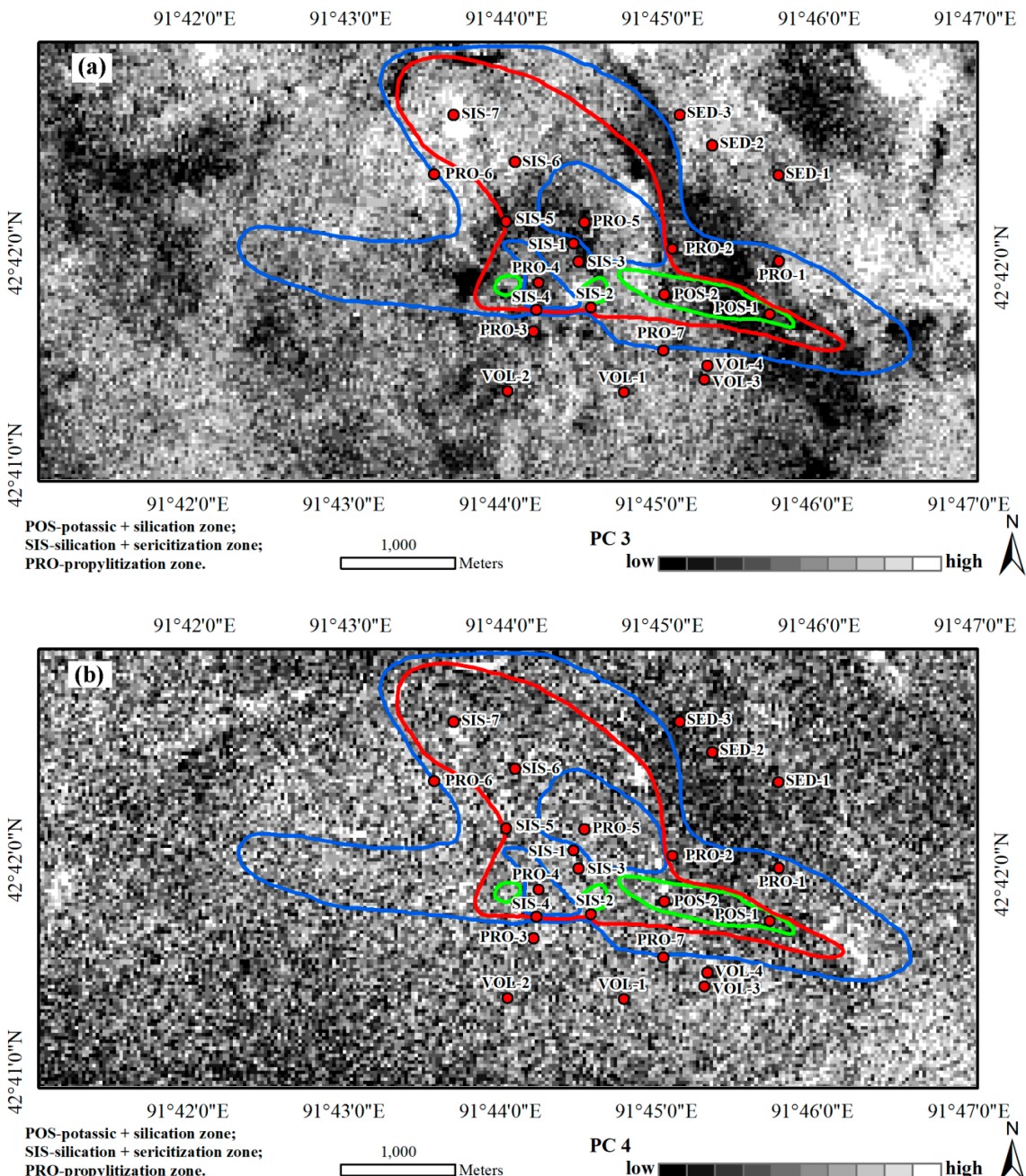

**Figure 4.** Principal component (PC) resultant images on ASTER (Advanced Spaceborne Thermal Emission and Reflection Radiometer) shortwave infrared (SWIR) bands of the Yudai Cu deposit. (**a**) PC3, (**b**) PC4.

## 3. Results

### 3.1. Absorption Feature Parameters Calculated Based on the Spectra of Hand-Specimen

The spectra of hand-specimen in the alteration zone resampled to ASTER bands were fitted using a quadratic polynomial function, from which the absorption feature parameters were calculated. In detail, the resampled bands 5, 6, and 7 were chosen to generate the absorption parameters of Al-OH-bearing alteration minerals. Table 5 shows the results that the sample had a positive quadratic polynomial with a wavelength position of maximum absorption within the 2167–2262 nm range, and the specific wavelength position and the depth of absorption feature are also presented. Similarly,

resampled bands 7, 8, and 9 were used to calculate the absorption parameters of Fe/Mg(OH)-bearing and carbonates alteration minerals, which has been illustrated in Table 5.

**Table 5.** The specific wavelength position and depth of absorption feature.

|  | Sample No. | Wavelength Position (nm) | Absorption Depth (%) |
|---|---|---|---|
| **Band 5/6/7** | POS-1 [1] | 2231 | 0.6362 |
|  | SIS-5 [2] | 2234 | 0.5766 |
|  | SIS-7 | 2225 | 0.5163 |
|  | PRO-2 [3] | 2238 | 0.9247 |
|  | PRO-6 | 2199 | 0.8626 |
| **Band 7/8/9** | POS-1 | 2380 | 0.6784 |
|  | POS-2 | 2340 | 0.6862 |
|  | SIS-1 | 2339 | 0.8042 |
|  | SIS-4 | 2359 | 0.8237 |
|  | PRO-3 | 2328 | 0.8094 |
|  | PRO-4 | 2355 | 0.9073 |
|  | PRO-5 | 2331 | 0.7561 |
|  | PRO-7 | 2321 | 0.8639 |

[1] POS-potassic + silication zone; [2] SIS-silication + sericitization zone; [3] PRO-propylitization zone.

According to Figures 3 and 4, the absorption features of Al(OH)-bearing, Fe/Mg(OH)-bearing, and carbonates alteration mineral could be found in different alteration zones, which were verified through the absorption feature parameters calculated based on the spectra of hand-specimen. Specifically, the largest depth of Al-OH maximum absorption feature appeared in propylitization alteration zone (PRO-2, PRO-6), followed by potassic + silication alteration zone (POS-1) and silication + sericitization alteration zone (SIS-7, SIS-5) (Table 5). For the Fe/Mg(OH) and carbonate absorption feature, the propylitization alteration zone (PRO-4, PRO-7) had the largest depth of maximum absorption feature, while the silication + sericitization alteration zone (SIS-4, SIS-1) and potassic + silication alteration zone (POS-2, POS-1) ranked the second and the third, respectively. However, the wavelength position of the maximum absorption feature did not present any pattern, according to Table 5.

### 3.2. Absorption Feature Parameters Estimated from ASTER Data for Mapping Hydrothermal Zoning Pattern

We fitted the quadratic polynomials using bands 5, 6, 7 and bands 7, 8, 9 of ASTER data to estimate absorption feature parameters for mapping hydrothermal zoning. Figure 5 shows the absorption feature parameters superimposed with the alteration zone and faults. Pixels with high value in the mapped depth of maximum absorption (Figure 5a,c) distributed around the ore body, which indicated the position of the ore body, although they were not completely consistent with the actual hydrothermal alteration zoning pattern mapped from the traditional method for identifying the alteration zone. The distribution of extracted alteration minerals information indicated that they were controlled by the faults.

Although the absorption feature parameters obtained with ASTER band 5/6/7 and band 7/8/9, respectively, appeared in a similar region, they presented different distribution characteristics. The mapped Al(OH) absorption feature depth gradually decreased from the center to the outside, while the depth of Fe/Mg(OH) and carbonates absorption feature reduced from north to south (Figure 5a,c). The results for wavelength position of Al(OH) absorption feature were scattered, while the wavelength position of Fe/Mg(OH) and carbonates absorption feature showed an encouraging amount of consistency with the absorption feature depth (Figure 5c,d). The wavelength position of Fe/Mg(OH) and carbonates absorption feature presented a redshift with the increase in the absorption depth.

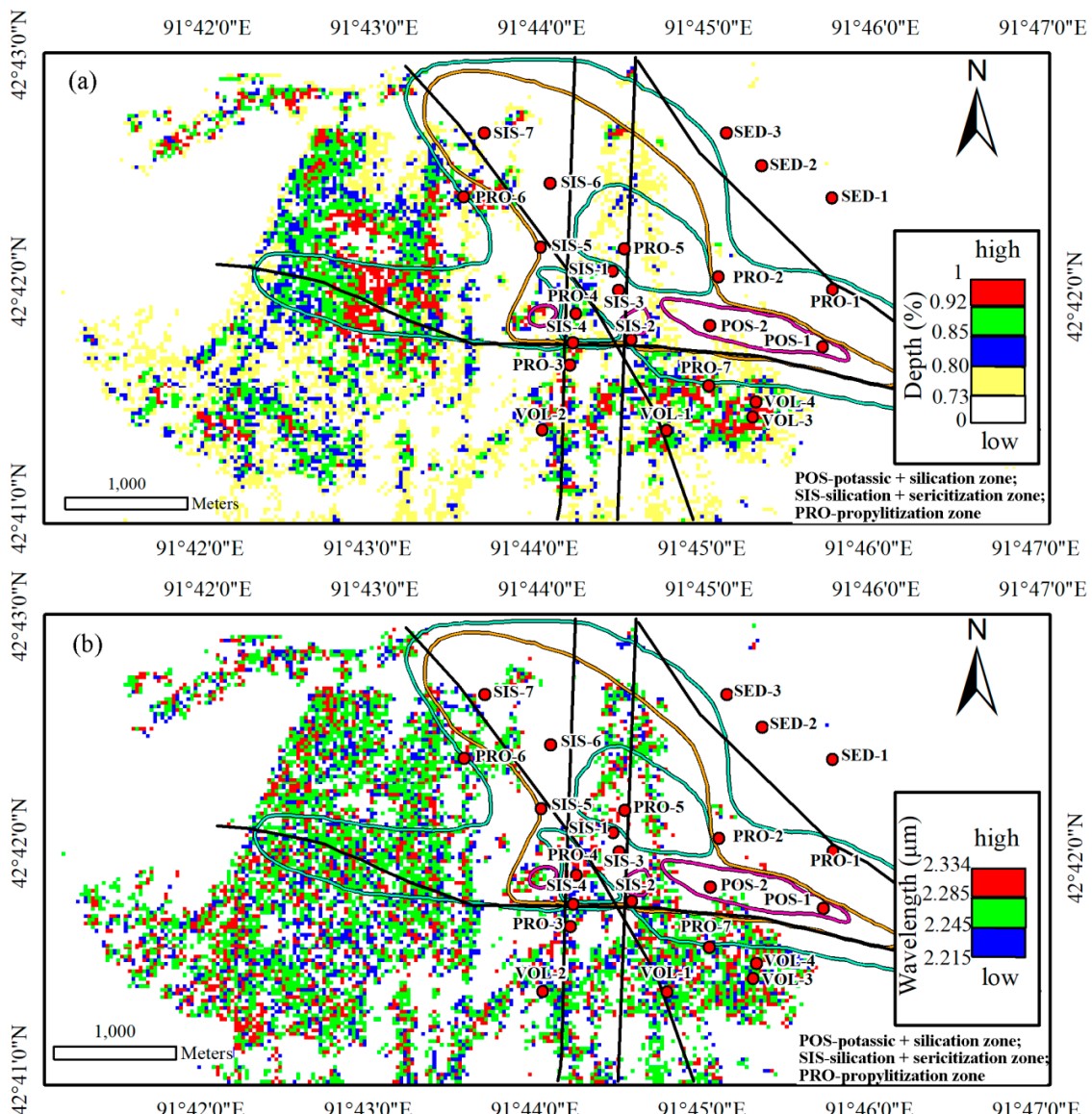

**Figure 5.** *Cont.*

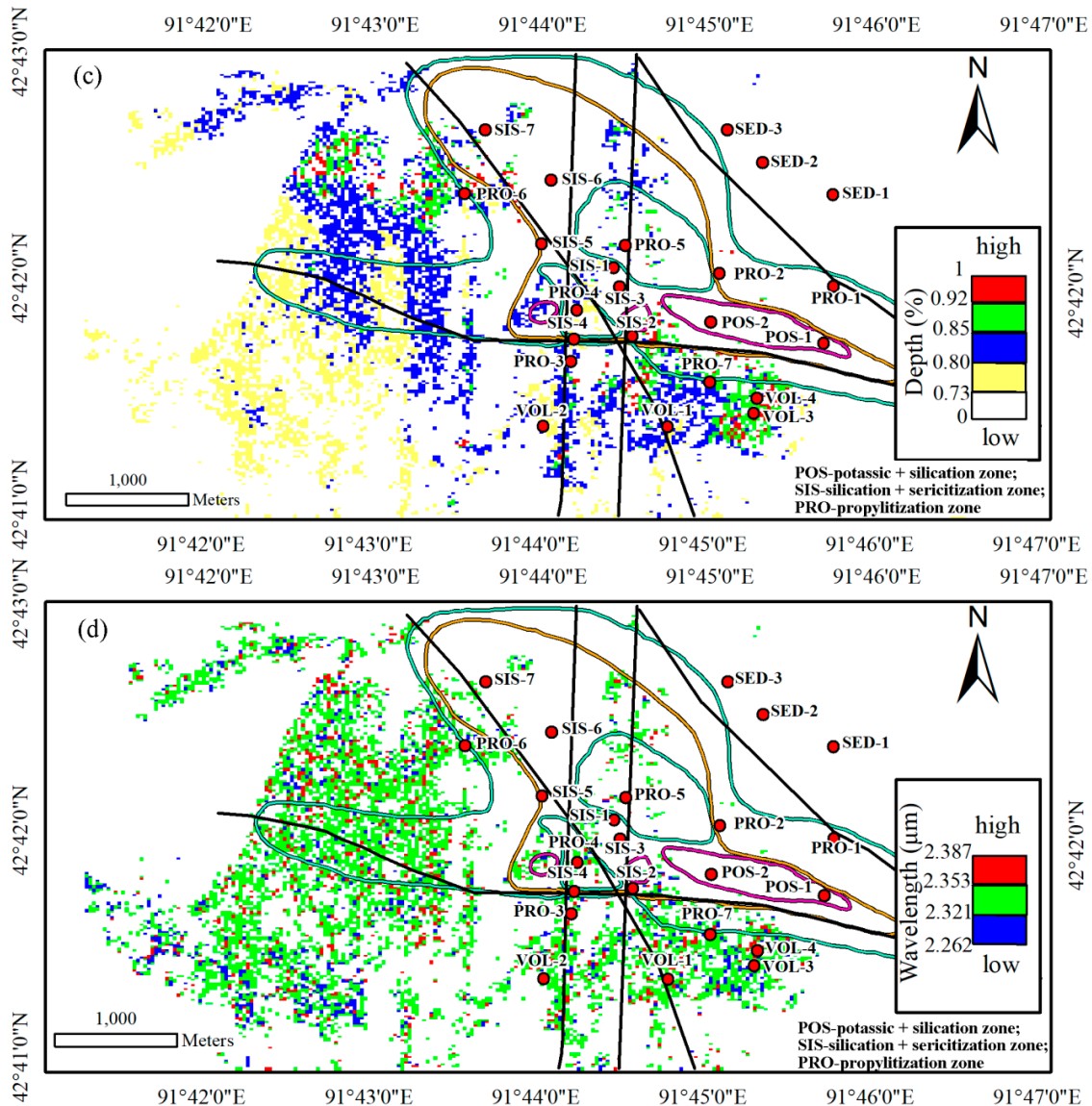

**Figure 5.** Mapping the absorption feature over the Yudai. Shown are the absorption feature parameters "absorption feature depth" (**a**,**c**) and the "wavelength of maximum absorption" (**b**,**d**) obtained with ASTER band 5/6/7 and band 7/8/9.

### 3.3. Validation of Mapped Pattern with the Ground Truths

Minerals contained with the hand-specimen were identified through the thin section, and the results are shown in Table 6. The photomicrographs of typical alteration minerals, such as sericite (Ser), epidote (Ep), chlorite (Chl), kaolinite (Kln), and calcite (Cal), are displayed in Figure 6.

The sample POS-1 had sericite and carbonate, and the absorption feature parameters of Al(OH) and carbonates were calculated based on the spectra of hand-specimen POS-1 (Table 5), which presented a good consistency. However, the estimated absorption feature parameters from ASTER data showed nothing at the location of POS-1 (Figure 5). Although the hand-specimen of PRO-3, PRO-4, PRO-5, and PRO-7 contained Al(OH)-bearing alteration minerals, the spectra of those hand-specimen measured by ASD could not be used to estimate absorption feature parameters of fundamental absorptions of Al(OH) (Table 5). However, those sampling points exhibited Al(OH) absorption features in the mapped hydrothermal zoning pattern using ASTER data (Figure 5a,b). For Fe/Mg(OH) and carbonates absorption features, there was a better correspondence between the results of petrographic thin section studies, and the absorption feature parameters were calculated based on the spectra of hand-specimen

and ASTER data (PRO-3, PRO-4, PRO-6, and PRO-7), where the Fe/Mg(OH)-bearing and carbonates alteration minerals could be identified using the above three methods at the same time.

**Table 6.** Major mineral composition of selected hand-specimens. Minerals were identified using petrographic thin section studies conducted under a transparent and polarized microscope.

| Sample No. | Petrography |
|---|---|
| POS-1 [1] | Feldspars, quartz, rock fragments, and alteration recrystallization into sericite and carbonate aggregates |
| PRO-3 [2] | Plagioclase crystals altered to sericite and kaolinite, epidotization and chloritization of rock fragments, vitreous volcanic dust alteration recrystallization into cryptocrystalline feldspar, chlorite and cryptolite aggregates |
| PRO-4 | Sericitization, kaolinization, and chloritization of plagioclase, amygdaloidal structure filled with quartz, chlorite, and carbonate |
| PRO-5 | Sericitization, kaolinization, epidotization, and chloritization of plagioclase, and epidotization and chloritization of opaques |
| PRO-6 | Rock fragments, crystal fragments, and volcanic dust alteration recrystallization into cryptocrystalline feldspar and chlorite |
| PRO-7 | Mild argillaceous of plagioclase, mild argillaceous of potassium feldspar, and chloritization of biotite |

[1] POS-potassic + silication zone; [2] PRO-propylitization zone.

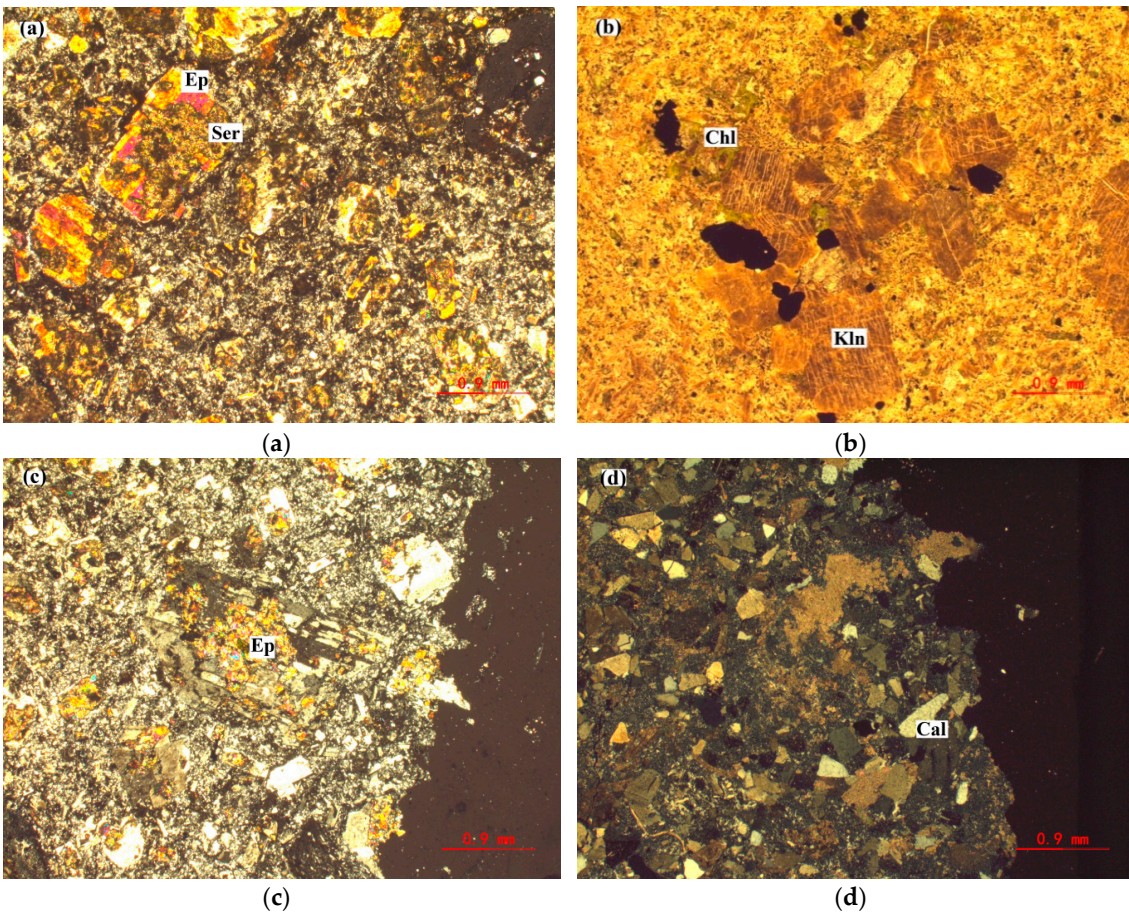

(a)   (b)

(c)   (d)

**Figure 6.** Photomicrographs: (**a**) Plagioclase crystals altered to sericite (Ser) and epidote (Ep) (in cross-polarized light). (**b**) Chlorite (Chl) produced by biotite alteration and kaolinization (Kln) of potassium feldspar (in plane-polarized light). (**c**) Plagioclase crystals altered to epidote (in cross-polarized light). (**d**) Alteration recrystallization into calcite (Cal) (in plane-polarized light).

## 4. Discussion

### 4.1. Performances of Absorption Feature Parameters

Compared with traditional statistical approaches like Relative Absorption Band Depth and Principal Component Analysis (Figures 3 and 4), the absorption feature parameters estimated from ASTER data were better in term of effectiveness on mapping the hydrothermal zoning pattern (Figure 5) in the study area. Previous studies on applying ASTER data to recognize hydrothermal alteration minerals relied more on other methods, such as Spectral Angle Mapper (SAM), Spectral Feature Fitting (SFF), Mixture-Tuned Matched-Filtering (MTMF), and so on [42,43]. Although Azizi et al. [42] had discriminated two main alteration zones, including propylitic and phyllic-argillic, using identified alteration mineral classes, the spatial positional relationship between the ore body and altered minerals could not have been expressed clearly. However, the band depth of absorption features of alteration minerals was found to be highly correlated with alteration mineral contents (intensity) [44], and mineralization should be closely related to alteration, while our results suggested that the pixels with high values in the mapped depth of maximum absorption corresponded to the ore body.

Nevertheless, we noted there was a deviation between the absorption feature parameters calculated based on the spectra of hand-specimen and those from the ASTER data. Some absorption features calculated based on the spectra of hand-specimen were not displayed at the corresponding positions in the hydrothermal zoning pattern mapped from ASTER data (POS-1), and even the opposite case occurred (SIS-2). However, compared with those for Al(OH) absorption feature, Fe/Mg(OH) and carbonate absorption features had a better consistency, although there remained a scaling issue between the spectral measurements made at ground and space.

The results obtained from the spectral feature fitting method [15] are shown in Table 7 to compare with the results resampled from United States Geological Survey (USGS) spectral libraries. The major alteration phases at Yudai were biotite (potassic alteration), sericite, chlorite, epidote, and carbonate, which were consistent with the results of spectral identification. Furthermore, the thin sections of the selected samples confirmed their occurrences in the study region. However, due to the limited spectral resolution of ASTER data, the fitted quadratic polynomial function was based on only three ASTER bands and, wasn't able to show the finer spectral features. As a result, the wavelength positions fitted by the quadratic polynomial functions were not too good for indicating specific alteration minerals. Although the wavelength position of Fe/Mg(OH) and carbonates absorption feature presented a redshift with the increase in the absorption depth, it was still uncertain to distinguish the mineral, even though Harald [24] had differentiated specific minerals, such as bronzite, goethite, jarosite, and hematite, using simulated Sentinel-2 visible and near infrared (VNIR) bands as input for estimating absorption feature parameters. The reasons for this discrepancy might be that the bandwidth of simulated Sentinel-2 VNIR bands [24] was narrower than those ASTER SWIR bandwidths, and the lower spectral resolution of ASTER SWIR bands made different absorption features inseparable. This could possibly be addressed by introducing separate band sets per target mineral to fit other functions in future research. With the support of the hyperspectral data, the subtle shifts of the wavelength position of a given absorption feature can be correlated to slight differences in chemistry, such as Al-Mg compositional changes in muscovite [45].

Several alteration minerals found in the hand-specimen could not have been extracted from the absorption feature parameters method in this study. For example, the spectra of the sample PRO-5 contained an absorption feature at around 2255, which indicated the existence of Fe/Mg(OH)-bearing and carbonate mineral. However, this alteration mineral could not have been identified from the absorption feature parameters mapped from ASTER 7/8/9. The field investigation is shown in Figure 7. The reason for this phenomenon could be the low spatial resolution of the ASTER data or alteration mineral contents.

**Table 7.** The rock spectra resampled to ASTER (Advanced Spaceborne Thermal Emission and Reflection Radiometer) band matches with United States Geological Survey (USGS) spectral library based on spectral feature fitting [41].

| Sample No. | USGS Matches | Score |
| --- | --- | --- |
| POS-1 [1] | muscovi3.spc, illite1.spc | 0.831, 0.821 |
| POS-2 | biotite.spc | 0.566 |
| SIS-1 [2] | talc2.spc, chlorit1.spc, dolomit1.spc | 0.597,0.547, 0.495 |
| SIS-2 | goethit1.spc, montmor6.spc | 1, 1 |
| SIS-3 | limonite.spc | 0.523 |
| SIS-4 | dolomit1.spc | 0.915 |
| SIS-5 | goethit1.spc, montmor6.spc, muscovid.spc | 0.991 |
| SIS-6 | goethit1.spc | 0.846 |
| SIS-7 | muscovi8.spc | 0.971 |
| PRO-1 [3] | jarosit1.spc | 0.842 |
| PRO-2 | unidentified | |
| PRO-3 | chlorit2.spc | 0.494 |
| PRO-4 | chlorit3.spc | 0.803 |
| PRO-5 | dolomit2.spc, epidote1.spc | 0.846, 0.666 |
| PRO-6 | kaolini1.spc, alunite5.spc, jarosit1.spc | 0.898, 0.862, 0.823 |
| PRO-7 | quartz3.spc | 0.913 |

[1] POS-potassic + silication zone; [2] SIS-silication + sericitization zone; [3] PRO-propylitization zone.

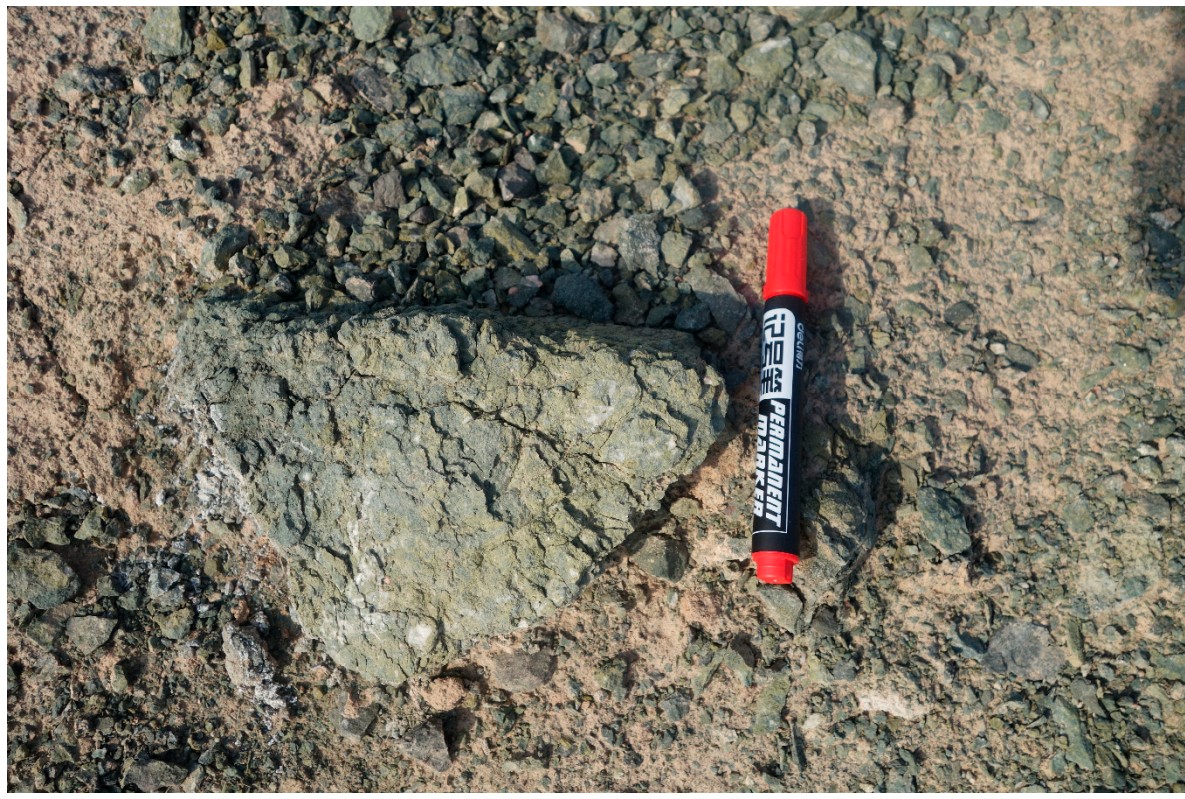

**Figure 7.** Field photographs at sampling point propylitization alteration zone -5 (PRO-5).

## 4.2. Application Prospects in Eastern Tianshan

The absorption feature parameters were used to map the hydrothermal alteration zoning pattern of the Yudai porphyry Cu deposit, where different alteration zones have similar alteration minerals, and the alteration zoning features are not obvious. This method could provide operational products

over larger areas to locate prospecting target of porphyry Cu deposit. Because the absorption feature parameters can contribute to locating hydrothermal zoning pattern, it expands the scope of application of ASTER data. This approach can be extrapolated to remote regions for exploring the new prospect of high-potential copper mineralization zones of the Kalatage copper polymetallic ore cluster area and other arid and semi-arid regions of the world with similar geology environments. There are many types of deposits in the Kalatage copper polymetallic ore cluster area, such as the South Meiling VMS deposit and Hongshi hydrothermal vein deposit. Thus, the method can further be used to map hydrothermal alteration zoning pattern of different metallogenic types, which is helpful for prospecting and exploration in eastern Tianshan.

## 5. Conclusions

ASTER SWIR data have been used to map the hydrothermal zoning pattern of a typical Cu deposit in western China based on absorption feature parameters. The results indicate that the mapped hydrothermal zoning pattern using the absorption feature parameters can effectively locate the position of the ore body, and are in good agreement with the geological mapping and field samples. Results in this study also indicate that the absorption feature parameters have great abilities to map hydrothermal zoning pattern of porphyry copper deposit under various conditions and can be extrapolated to different regions for exploring similar copper mineralization zones. We foresee wide applications in the future.

**Author Contributions:** Conceptualization, M.W. and Q.W.; Data curation, M.W.; Formal analysis, M.W.; Funding acquisition, K.Z.; Investigation, M.W.; Methodology, M.W., Q.W., and J.W.; Project administration, K.Z.; Resources, J.W.; Software, K.Z.; Supervision, Q.W. and J.W.; Validation, M.W. and J.W.; Visualization, M.W. and J.W.; Writing—original draft, M.W.; Writing—review and editing, Q.W. and J.W.

**Funding:** This research was funded by The belt and Road Team of Chinese Academy of Sciences, grant number 2017-XBZG-BR-002, National Natural Science Foundation of China, grant number U1803117, and Bulletin of Chinese Academy of Sciences, grant number XDA19030204.

**Acknowledgments:** We thank Professional Laboratory for Geochemical Analysis and Testing, Xinjiang Institute of Ecology and Geography, CAS for providing the microscope to carry out the identification of thin sections.

**Conflicts of Interest:** The authors declare no conflict of interest.

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
