# Peer review of "Mapping Hydrothermal Zoning Pattern of Porphyry Cu Deposit Using Absorption Feature Parameters Calculated from ASTER Data"

_remotesensing, doi:10.3390/rs11141729_

Round 1

Reviewer 1 Report

This study is a very straightforward implementation of standard spectral feature absorption approaches.

The study would be strengthened with the integration of other ASTER image datasets covering the VNIR and TIR waveranges and also with the inclusion of other spatial datasets such as topography, land cover etc. Integration of additional sensor datasets with complimentary spectral and spatial attributes would also add some additional insights into the geological setting and distribution of the altered rocks. The spectral analysis methods are very basic and don't address a number of the key issues affecting remote mapping of mineral deposits such as mixed pixels. Overall there is nothing new in this study just a very basic geological analysis of ASTER data over a porphyry deposit

Reviewer 2 Report

Many congratulations to the authors for having completed a nice and sound research work. A minor revision is needed before the manuscript could be accepted for publication. I have marked my comments in the manuscript. Overall, very good effort.

Reviewer 3 Report

Brief Summary

The authors demonstrate that calculated ASTER absorption features can locate Cu porphyry hydrothermal zones and posit that the technique can be used to prospect for minerals in remote locations.  The curve-fitting technique provides an approach that utilizes low-spectral resolution imagery for small-scale (large spatial extent) mapping.  The possibility of locating faults associated with the Cu porphyry mineralization is an interesting speculation.

Broad Comments

The manuscript’s strength is the use of satellite remote sensing as well as field and laboratory work.  Testing the use of low-spectral resolution ASTER data for reconnaissance mapping or mineral exploration over large areas is to be commended.  In addition, discussing multiple techniques for identifying alteration zones is a contribution that prompted a question about other interpolation techniques.  The main weakness of the manuscript is the use of English to explain the work accomplished.  At times, it was difficult to follow the manuscript because of the words used.

Specific Comments

Line 18: Large scale? Large scale applies to small areas and whereas small scale applies to large areas usually covered by satellite imagery.  Change “large scale information” to “small scale information”.  Also, change “developing” to “developments”.

Line 43: Change “Since” to “Because”. The word “since” relates to time after an event.

Lines 44-45: Sentences unclear.

Lines 50-59: Providing some traditional information about major element geochemistry of unaltered rocks and minerals in the Yudai copper deposit in the Kalatag district may strengthen the authors’ argument.

Line 126: By changing “are spotting” to “occur” the reader easily can comprehend the thought.

Line 147: Change “cloud amount” to “cloud cover”.  Cloud cover is normally used when describing the portion of the image that is free of clouds.

Line 175: Is 5 the correct number of reflectance spectra? Clarify sentence.

Lines 176-177: Sentence is unclear.  The use of the verb “were focused” does not make sense.

Lines 181-182: Not a sentence.  If the word “ While” is eliminated then there is a sentence.

Line 186-189: Perhaps the geochemical or petrographic information on the hand specimens could provide insight about the absorption features in hand specimens versus the 2350 nm absorption feature noted in all alteration zones.

Line 215-216: Were “all samples” the number of samples listed in Table 2. If so, then state “all 23? samples”.

Line 277: Add the word “scale” after “ 1: 50,000”.

Lines 301-302: Sentence is unclear. By “rules” do you mean that there is no pattern or generalization?

Lines 311-312: Last sentence of paragraph is not necessary as the previous sentence mentions the fault control of alteration.

Line 317: Changing the words “present a mess” to “ are scattered” might better convey the distribution of results.

Table 6: For sample POS-1, the term ”vitrinite” refers to a type of kerogen.  Perhaps, the term “vitreous” which refers to glassy fragments is more appropriate.  Glass shards or altered glassy fragments are not uncommon in altered volcanic rocks. For PRO-4, I do not know what is meant by an “almond stone”.  For other petrographic samples, perhaps using the preposition “of” with the words chloritization or epidotization would help the reader.  For example, “chloritization of biotite” or ”kaolnization of plagioclase”.

Line 337: Nice to see the labeled photomicrographs!

Line 408: Delete the word “As” and begin the sentence with the word “Thus”.

Line 414: Changing the words” geological profile” to geologic mapping” might better convey what was done.

Line 415: Change “has” to “have”,

Line 417: Delete the word “its”.

Reviewer 4 Report

Please see comments in provided PDF after an extensive review of the research work. Despite the interesting approach and work, there are a few major issues the authors should revisit and try to solidify the context.

In particular, I am raising some concerns on speculative arguments connecting observations (that are not accurately described on a mathematical basis) with conclusions. Visual inspection that has a 30m resolution over a 60x60 km2 region is not sufficient by itself to provide a strong argument on how things are detected remotely to the extent described. The authors should provide a more rigorous description of the sensing protocols and related details.

I would welcome a second round of review in that sense.

Round 2

Reviewer 4 Report

Dear Authors, thank you for the detailed response to all the comments and points raised during the first round of reviewing. I have gone exhaustively through the revised version you provided and I have made some new remarks in the PDF file I am resubmitting with my new evaluation.

I would like to stress two facts:

- The quadratic function in Eq. (1) is used to interpolate across bands. However, no coefficients and their related errors in the interpolation are provided in the text, neither a correlation coefficient or chi-square value is given to have an idea about the quality of the fit.

- The technique is clearly capable of getting some results on characterizing the ore deposit, however, in general, it suffers some weaknesses. Along these lines, I would urge you to explicitly state existing limitations in regards with the conclusions you are drawing on its effectiveness, as well as potential improvement when combined with other techniques (give example(s) and references)

These are the two things I consider important before I provide a final positive evaluation on the research work you have conducted over the years. Some minor typos and comments should also be looked upon carefully. Thank you for your collaboration.
